# Machine learning for the detection and diagnosis of cognitive impairment in Parkinson's Disease: A systematic review

**Callum Altham** *, **Huaizhong Zhang**, **Ella Pereira**

Department of Computer Science, Edge Hill University, Ormskirk, Lancashire, United Kingdom

☯ These authors contributed equally to this work.
\* althamc@edgehill.ac.uk

## Abstract

**Data Availability Statement:** All relevant data are within the manuscript and its Supporting information files.

**Funding:** The author(s) received no specific funding for this work.

### Background

Parkinson's Disease is the second most common neurological disease in over 60s. Cognitive impairment is a major clinical symptom, with risk of severe dysfunction up to 20 years post-diagnosis. Processes for detection and diagnosis of cognitive impairments are not sufficient to predict decline at an early stage for significant impact. Ageing populations, neurologist shortages and subjective interpretations reduce the effectiveness of decisions and diagnoses. Researchers are now utilising machine learning for detection and diagnosis of cognitive impairment based on symptom presentation and clinical investigation. This work aims to provide an overview of published studies applying machine learning to detecting and diagnosing cognitive impairment, evaluate the feasibility of implemented methods, their impacts, and provide suitable recommendations for methods, modalities and outcomes.

### Methods

To provide an overview of the machine learning techniques, data sources and modalities used for detection and diagnosis of cognitive impairment in Parkinson's Disease, we conducted a review of studies published on the PubMed, IEEE Xplore, Scopus and ScienceDirect databases. 70 studies were included in this review, with the most relevant information extracted from each. From each study, strategy, modalities, sources, methods and outcomes were extracted.

### Results

Literatures demonstrate that machine learning techniques have potential to provide considerable insight into investigation of cognitive impairment in Parkinson's Disease. Our review demonstrates the versatility of machine learning in analysing a wide range of different modalities for the detection and diagnosis of cognitive impairment in Parkinson's Disease, including imaging, EEG, speech and more, yielding notable diagnostic accuracy.

**Competing interests:** The authors have declared that no competing interests exist.

## Conclusions

Machine learning based interventions have the potential to glean meaningful insight from data, and may offer non-invasive means of enhancing cognitive impairment assessment, providing clear and formidable potential for implementation of machine learning into clinical practice.

## Introduction

Parkinson's Disease (PD) is the most common neurodegenerative disorder [1], characterised by motor and non-motor symptoms including dyskinesia, tremors and balance issues [2]. Over 145K people in the UK are estimated to be living with PD [3], making PD the second most common neurological disease in individuals over the age of 60. PD has an estimated global prevalence rate of 1%, doubling global PD populations between 1990 and 2016 making PD the fastest-growing neurodegenerative condition in the world [4–6].

PD sufferers have a higher risk of developing severe cognitive complications resulting in consistent and damaging cognitive impairments (CI) giving rise to a noticeable loss in cognitive functioning and behavioural abilities, and can lead to the development of overall cognitive decline characteristic of dementia, known as Parkinson's Disease Dementia (PDD) [7]. Between 70–95% of PD patients are likely to experience some degree of CI as PD advances, with PDD frequently developing 10–20 years post diagnosis [8, 9], with potential for severe impacts on overall quality of life, familial relationships and societal functioning [10]. Treatment and disease management create severe burdens at medical [11], economic [12, 13] and personal levels, with identification of a direct, specific cause for CI development remaining a working, disputed research area [14, 15]. Presentation of CI is complex and diverse, occurring across a number of cognitive domains [16], including visuospatial [17], working memory [18] and psycho-motor speed [19]. Patients also experience widespread variations in onset, severity and progression [16], with diagnosis commonly carried out using clinician-led assessments of cognitive and processing ability to identify at least one dementia syndrome within established PD [8].

Several assessments are available to assess the entire spectrum of cognitive abilities, including the Benton Judgement of Line Orientation (JoLO) [17], Letter-Number Sequencing Task (LNST) [18], Symbol Digit Modalities Test (SDMT) [19] and Montreal Cognitive Assessment (MoCA) [20]. However, some studies consider such assessments limited since they only identify cognitive decline once symptom presentation has begun [20]. Therefore, research is beginning to focus on analysis of additional data modalities including gait analysis [21–23], functional connectomics [24], electroencephalogram [25, 26], amyloid PET [27], FDG-PET [28, 29], and quantitative susceptibility mapping [30–34]. However, diagnosis is still reliant on clinical features and standardised clinical criteria including the UK Parkinson's Disease Society Brain Bank (UKPDSBB) [35]. Such criteria rely largely on expertise and knowledge of a neurologist, however they can still be unreliable, with diagnostic accuracy assumed to be just over 80% in specialised neurology centres [36].

Machine learning (ML) techniques are increasingly being used within the healthcare industry for a wide range of tasks. Publications using ML for detection and diagnosis of CI have increased to investigate the potential uses for these techniques in attempt to mitigate these limitations and provide additional measures that may potentially identify CI in a quicker and earlier manner. Such techniques enable systems to learn by example by studying large datasets

and extracting meaningful representations [37] which are then used to make decisions based on learned information without explicit programming [38]. Using ML to analyse CI in PD is yet to be fully analysed and validated despite this growing usage and expansion in literature. This work aims to provide an overview of published studies applying ML to detecting and diagnosing CI, evaluate the feasibility of implemented methods, their impacts, and provide suitable recommendations for methods, modalities and outcomes. We aim to provide an overview that functions as a starting point for further, more detailed analyses of CI in PD, influencing research into identification of early stage, non-invasive markers of disease progression. This has the potential to allow for detection and diagnosis at the earliest stages, allowing much needed intervention and preventative care that could slow overall disease progression [39, 40].

The paper is structured into a number of separate sections. Firstly, a background provides context on PD, CI, and ML's role in detection and diagnosis. Review protocol outlines the systematic literature review process. Observations and findings analyses literature characteristics and explores ML's application in CI detection, emphasising performance across data modalities. Discussion considers the implications of findings within the context of PD and CI, fostering critical analysis and integration of ML insights into clinical practice. Finally, this paper is concluded in conclusions.

## Background

### Parkinson's Disease

PD is a progressive neurodegenerative disorder primarily affecting the motor system [1], characterised by the extensive loss of dopaminergic neurons in the substantia nigra pars compacta [41] alongside pathological processes, including aggregation of $\alpha$-synuclein protein [42], mitochondrial dysfunction [43], oxidative stress [44] and neuroinflammation [45]. This region of the brain is integral in control of motor functions, and deterioration of this region results in the hallmark symptoms of PD. As PD progresses and develops, up to 50% of these crucial neurons are lost at the point of symptom presentation, significantly reducing dopamine levels. This depletion of dopamine results in a primary manifestation of pronounced motor symptoms including resting tremors, bradykinesia, muscle rigidity and postural instability [6], underscoring the fundamental nature of PD as a motor disorder.

Despite PD being a fundamentally motor disease, it also encompasses a wide variety of non-motor symptoms that can cause impacts to a patient's quality of life including anosmia (loss of smell) [46], sialorrhea (excessive salivation), difficulties with speech and swallowing [47] alongside changes in vision and hearing [48, 49]. Additionally, dopamine is also involved in the regulation of cognitive processes [50, 51], resulting in PD affecting executive functions, attention, visuospatial and language skills. These cognitive changes typically become more pronounced as the disease advances and are crucial for a comprehensive understanding of the disease alongside traditional motor symptoms.

**Cognitive impairment.** Most notably, CI has begun to show prominence as a non-motor symptom of PD [10]. Alongside dopamine depletion, contributions to cognitive dysfunction are made by accumulation of Lewy bodies and Lewy neurites in the cerebral cortex, limbic system and other brain areas [52, 53]. Similarly, cholinergic [54], serotonergic [55], and noradrenergic [56] systems have been implicated for involvement in development of CI in PD.

CI is a complex condition causing profound impacts on daily life and well-being [10]. CI encompasses a wide range of cognitive deficits extending beyond the expectation of memory impairment, including executive dysfunction [57], attention [58], visuospatial impairment [59], language difficulties [60], memory problems [57, 61], and mood disturbances such as anxiety and depression. Occurrence of CI is heterogeneous, with variations in patterns and

severity. Differentiating between levels of severity [62], including Normal Cognition (PD-NC), Mild Cognitive Impairment (PD-MCI), and PD-Dementia (PDD) is crucial, with the latter constituting a severe and pervasive deficit characterised by significant impairments to daily functioning [7].

Diagnosis involves a comprehensive assessment battery, integrating elements including clinical evaluations, neuroimaging, and neuropsychological testing. A key framework for this is the Movement Disorder Society Criteria (MDS), which provides thorough evaluation of cognitive domains including attention, memory, and executive functioning [62]. A number of assessment and screening tools can be used, including the Mini-Mental State Examination (MMSE), which assesses general cognitive ability [63], and the MoCA, offering a more detailed evaluation of cognitive function [20]. The MoCA is generally considered to have a higher diagnostic power compared to the MMSE due to its broader cognitive domains used in assessment, heightened sensitivity to MCI and early dementia, and better adjustment for education levels, making it a superior tool in detecting subtle cognitive changes and must be used for cognitive screening of PD in clinical practice. Neuroimaging techniques play a considerable role through identification of structural and functional changes in the brain associated with continuing cognitive decline [64]. Such tools allow clinicians and researchers to use standardised, comprehensive approaches to the diagnosis of CI in PD, potentially allowing for earlier detection and management of this aspect of the overall PD condition.

## Machine learning

ML has emerged as a valuable tool in a variety of healthcare applications, including the detection of PD [65, 66] and a number of related memory disorders [67, 68], proving itself worthy of consideration for analysing CI. ML encompasses various approaches including supervised, unsupervised, deep and ensemble learning. Therefore, the following section provides an overview of the required theory to understand the wide array of ML methods that have been implemented in discovered studies, with the applications of these techniques discussed further on.

**Supervised learning.** Supervised learning involves algorithms trained to make decisions based on a set of labelled examples. A dataset of input data and their expected output labels are used to train the model [69] with internal parameters adjusting to minimise differences between predictions and expected labels. The trained models then generalise knowledge to make predictions on new data to solve either regression or classification tasks. Supervised learning methods cover a large wealth of model types encompassing both classification and regression tasks.

Classification methods are fundamental ML methods that are used to assign categorical labels to data points based on their provided input features [70]. These methods are vital in their ability to differentiate and categorise provided data into distinct classes, enabling models to recognise patterns and perform decision making [71]. Classification models perform a variety of approaches, from simple binary classifiers to more complex multi-class systems, each of which are designed to address a specific type of classification problem.

Regression methods are essential tools in ML that allow for the prediction of continuous numerous variables based on a pre-determined set of input features [72]. These methods are widely utilised due to their ability to effectively model relationships between variables, understand patterns in data and make accurate predictions [73]. Regression models vary widely in terms of technique, and therefore are typically tailored for use in a specific scenario. Tree based ML methods cover a class of algorithms used widely for classification and regression tasks [74] including Decision Trees (DT), Random Forests (RF) and Gradient Boosting Trees (GBT). Favoured due to their simplicity, interpretability and proficiency in handling

structured data [75]. At their core, these methods work by recursively splitting datasets into subsets based on input features, creating a hierarchical tree like structure, with nodes representing a particular rule and branches denoting outcomes.

Support Vector Machines (SVMs) are a supervised ML algorithm typically used for classification or regression. SVMs excel at binary classification tasks but also have the potential for adaptation for multi-class classification and regression [71]. The principle concept of SVMs relies on the finding of a 'hyperplane', or decision boundary, that is able to effectively separate the different classes of data points. This decision boundary is positioned to maximise the distance between the decision boundary and the closest data points of each respective class or 'support vectors' to ensure effective categorisation [76].

Linear Regression (LiR) methods assume the presence of a linear relationship between input features and an expected target variable, and therefore are more suited for standard linear relationships [72]. Logistic Regression (LoR) is used to model the probability of a binary outcome, using an 'S'-shaped logistic function that maps any real-valued number into the range 0 to 1, making it particularly suitable for binary classification tasks [77]. Polynomial Regression (PR) extends this ability by allowing for the analysis of features with a polynomial relationship to the target variable [78].

Neighbour-based methods are a group of ML techniques that rely on the concept of similar data points sharing common characteristics, allowing predictions or recommendations to be made based on the proximity of data points to one another. The most commonly utilised neighbour method is that of the K-Nearest Neighbour (K-NN) method. The K-NN algorithm is focused on the concept of finding the K-nearest data points to a given target point. It then makes predictions based on the majority class or the average value of these K-nearest neighbours [79] respectively for classification and regression tasks.

Naïve Bayes (NB) is a ML algorithm primarily used for classification tasks with predefined classes or categories. The algorithm operates on the basis of Bayes theorem, in which the probability of an item belonging to a specific class is calculated based on the observed features [70]. 'Naïve' identifies an assumption made during the modelling process, in which all features used for classification are assumed to be independent of each other in producing the class label [80].

Discriminant Analysis (DA) is a fundamental technique in ML aimed at simplifying complex datasets by reducing the number of features or variables involved whilst retaining crucial information [81]. This approach is vital for addressing challenges associated with high-dimensional data. Notable supervised methods for discriminant analysis are Linear Discriminant Analysis (LDA) and Quadratic Discriminant Analysis (QDA).

Genetic Programming (GP) is a ML approach inspired by the mechanisms of natural selection and evolution [82]. At its core, GP emulates the process of biological evolution to automatically create and refine computer programs to tackle complex problems. GP processes begin with a population of randomly generated computer programs, often represented as trees or graphs [83].

Over multiple generations, GP continues to evolve programs, gradually improving their ability to solve the problem and moving closer to optimal or near-optimal solutions [82].

Hybrid ML methods combine the strengths of multiple ML techniques to address complex and diverse problem domains more effectively. These methods often integrate both traditional statistical approaches and modern deep learning algorithms [84]. Hybrid models are particularly valuable when dealing with multifaceted data types or when a single ML technique may not capture all the nuances of a problem.

**Unsupervised learning.** Unsupervised learning is an alternative method of ML technique in which algorithms find patterns, rules or structures in unlabelled data. No explicit labels are provided alongside training data. Instead, algorithms are expected to uncover all relationships,

groupings and representations within the data to group data into a set of categories or 'clusters' based on common features and patterns [85]. Common unsupervised techniques include clustering, in which algorithms group items together based on similar data points, and dimensionality reduction, which aims to simplify complex data by representing it in a lower dimensional space.

Dimensionality Reduction (DR) is fundamental in ML, and involves reducing the number of input features while maintaining important information to improve the efficiency and effectiveness of algorithms in handling high-dimensional data [86]. Notable unsupervised methods for DR are Principal Component Analysis (PCA), and Non-Negative Matrix Factorisation (NMF). PCA is a technique used for understanding the structure of high-dimensional data by reducing data dimensions without the loss of significant information by focusing on capturing the maximum variance present in data by identifying combinations of features called principal components [87] which often reveal underlying patterns in the data. NMF is a technique that decomposes a given data matrix into two or more matrices, where all the numbers in these matrices are non-negative. These matrices capture underlying patterns and relationships within the data, allowing us to represent the original data as a combination of these patterns, which can be easier to interpret and analyse [88].

Clustering is an essential unsupervised ML technique that groups data points based on their inherent similarities, revealing hidden structures within data [89]. Two prominent clustering methods are K-Means Clustering (KMC) and Gaussian Mixture Models (GMM). KMC aims to partition data into K clusters, where each data point is assigned to the cluster with the nearest mean (centroid). K-Means is computationally efficient and suitable for scenarios with roughly spherical and equally sized clusters [90, 91]. GMM, on the other hand, models data as a mixture of multiple Gaussian distributions, offering greater flexibility in handling clusters with varying shapes, sizes, and densities. It employs the Expectation-Maximisation (EM) algorithm to iteratively optimise its parameters [92].

**Deep learning.** Deep Learning (DL) is a specialised subset of ML focused on the training of Artificial Neural Networks (ANNs), which are models inspired by the structure of the brain [93]. DL differs significantly from traditional ML methodologies by utilising 'Deep' Neural Networks (DNNs), which are characterised by an interconnected, layered network architecture. The term 'deep' stems from the advanced capability of the network to automatically extract and learn features from raw data, bypassing the requirement for traditional, pre-defined, non-trainable feature extractor blocks [94]. This direct extraction of hierarchically organised, trainable features enables these models to perform complex pattern recognition and decision-making processes in a more effective manner. Each layer within a DNN is comprised of a number of interconnected nodes or neurones, that are capable of sequentially processing and transforming data, creating a hierarchical, structured representation of the input, significantly enhancing the models ability to learn from vast amounts of data and make informed predictions and decisions. Convolutional Neural Networks (CNNs) represent a specialised form of DNNs designed for processing grid-like data, such as images and videos [95]. CNNs have significantly advanced computer vision tasks by employing convolutional layers to apply filters (kernels) to input data, capturing local patterns. Pooling layers reduce spatial dimensions, and fully connected layers facilitate classification or regression [96]. CNNs dominate fields like image classification, object detection, facial recognition, and image generation [29]. However, it comes with challenges such as the need for datasets containing large numbers of labelled samples, with common deep learning datasets such as the ImageNet dataset including over 3.2 million samples [97], the risk of overfitting in deep networks, and interpretability concerns in complex models.

**Ensemble learning.** Ensemble learning is a ML technique focusing on combining multiple individual models to produce a more coherent and formidable 'ensemble' model [98]. Such techniques are based on the underlying idea that by aggregating predictions or decisions from a wide variety of models, the overall decision making performance can be considerably improved compared to using a singular model and any minor issues in model architectures can be mitigated [99]. Ensemble learning is applied in a wide range of ML tasks, including classification, regression and anomaly detection. These techniques are increasingly valuable when faced with datasets that are complex or noisy, as the diversity of the models allows for the improving of overall performance despite this noise. Similarly, with ML models consistently impacted by the effects of issues such as overfitting when a model learns the training data too well and struggles to perform on fresh data, ensemble methods can mitigate the impact overfitting may have on a model by improving the overall resilience and generalisation of the model [100].

**Model choice.** ML techniques are becoming widely used in healthcare, as detailed above, and therefore have considerable potential for use in detecting CI in PD. However, considerations still need to be made to implement such techniques. Choosing a suitable model depends largely on the nature of the data, problem complexity, and available resources [37]. DL models show great promise for use due to their ability in capturing intricate patterns in larger datasets [101]. However, traditional models including K-NN, SVM and RF still have potential to be effective in a number of situations, particularly when the presence of labelled data is limited and data interpretation is crucial [102]. An overview of all ML models used in the discovered papers are shown in Table 1.

## Review protocol

No registered protocol exists for this review.

## Purpose of review

The aim of this systematic review is to determine if ML approaches are effective for detection and diagnosis of CI in PD, and identify key methodologies, algorithms, and performance

**Table 1. Commonly used ML applications for CI detection and diagnosis in PD.**

| Algorithm | Description |
|---|---|
| DT | Hierarchically splits data, creating a tree-like structure with decision nodes and leaf nodes. |
| LiR | Models relationships between dependent variable and independent variables by fitting a straight line to the data |
| LoR | Models relationships between dependent variable and independent variables using a logistic function |
| SVM | Finds a hyperplane in a high-dimensional space to best separate data points into distinct classes. |
| NB | A probabilistic classifier based on Bayes' theorem. It assumes that features are conditionally independent |
| KNN | Assigns a data point to the majority class among its K nearest neighbours based on a distance metric |
| CNN | Employ convolutional layers to capture local patterns and spatial relationships in data. |
| RF | Combines multiple decision trees, with each tree voting independently towards the final decision |
| Hybrid | Combine two or more different machine learning approaches to leverage the strengths of each method. |
| Clustering | Applied to group similar data points, aiding in data segmentation and pattern discovery |
| DR | Reduces the complexity of high-dimensional data, making it more manageable for analysis and visualisation. |
| DA | Maximises class separability by finding linear combinations of features. |
| GP | An evolutionary algorithm that evolves computer programs to solve complex problems. |

metrics. Whilst a number of reviews have previously been conducted to determine the feasibility of ML techniques for the detection of PD alone [103–105], systematic reviews into the area of CI detection for PD are limited, with the most recent review found being published in 2022 [106], containing only half the number of papers in this review. Additionally, the areas of ML and AI are a constantly evolving and changing area, with newer and more advanced techniques becoming frequently available, and as a result it is important to keep abreast of all techniques that are currently being used within this area. Therefore this review expands on previous reviews to include the most recent papers in what is a growing and emerging research area. Based on this and following the PICO (Patient/Population, Intervention, Comparison, Outcomes) guidelines for review question formation [107], by conducting this review, we aim to discuss and answer the following question:

In patients with Parkinson's Disease, how effectively does machine learning-based detection and diagnosis differentiate cognitive impairment from normal cognition?

### Search methodology

A number of systematic review methodologies exist, including AMSTAR [108], PICO [107] and Cochrane [109], however, the PRISMA (Preferred Reporting Items for Systematic Reviews and Meta-Analyses) methodology [110, 111] is a well established, widely recognised approach for systematic reviews and meta-analyses, and is commonly used in both medical and computational based research since it provides a structured and transparent framework for literature searches, study selection and reporting findings, therefore, this work was based on PRISMA guidelines [110, 111]. Fig 1 provides the search, screening, eligibility and extraction steps carried out in this review.

### Literature sources

Whilst this review is focused on ML techniques for a particular research area, it must be considered how this research area is one of a largely medical nature rather than solely computational. Therefore, there is a need to consider sources from both medical and computational viewpoints whilst simultaneously considering sources of a generic nature that may cover any missed topic areas. In this work, we consider four databases spanning computational, medical and generic scientific groups: PubMed (pubmed.ncbi.nlm.nih.gov), IEEE Xplore (ieeexplore.ieee.org), Scopus (scopus.com), and ScienceDirect (sciencedirect.com).

### Search strategy

To retrieve all relevant literature, two sets of search terms were chosen for use in searching the aforementioned databases, including a primary set of terms: (1) Parkinson, (2) cognitive impairment, (3) machine learning, (4) deep learning, (5) diagnosis, (6) detection, (7) classification, and (8) identification. A set of secondary terms were also used interchangeably with primary keywords (1)-(3) to discover additional results. These search terms were combined with Boolean operators to produce search strings tailored to each database listed in S1 File. These search strings were then varied accordingly utilising the secondary search terms listed in S2 File. A comprehensive literature search was conducted on the PubMed, IEEE Xplore, Scopus and ScienceDirect databases with no restrictions and publishing dates from the beginning of the database to February 2024, with a search conducted on the 25th February 2024, resulting in a total of 1,052 available results.

**Fig 1. PRISMA flow diagram of the literature search, screening and extraction procedures for inclusion.**

## Inclusion and exclusion criteria

Based on the review objectives above, inclusion and exclusion criteria were created to ensure all literature align with these objectives and provide an effective overview of the scope of the research area. Therefore, for inclusion in this review, studies need to satisfy at least one of the following criteria:

(a) Classification of Cognitive Impairment (PD-CI), Mild Cognitive Impairment (PD-MCI), and Parkinson's Disease Dementia (PDD) from Normal Cognition (PD-NC)

(b) Classification of PD-CI, PD-MCI and PDD from other memory-based disorders (e.g. Alzheimer's Disease (AD) or Dementia with Lewy Bodies (DLB))

(c) Prediction of conversion from PD-CI/PD-MCI to PDD

(d) Prediction of future cognitive assessment scores

(e) Identification of biomarkers for the development of CI in PD sufferers

Studies that met any of the following exclusion criteria were not chosen for inclusion in this review:

(a) Studies investigating CI present before the onset of PD symptoms

(b) Studies not conducted on human participants or secondary data gathered from humans

(c) Studies focusing on the analysis of symptoms that do not include cognitive symptoms

(d) Studies providing a limited or insufficient description of data modalities, subjects or ML methods utilised

(e) Studies conducted in a language other than English

## Data extraction

Each paper gathered from sources mentioned in Literature Sources had identical information extracted, with this information included in S3 File in the supplementary material:

(a) Publication Year

(b) Data Source

(c) Activity Type (diagnosis, differential diagnosis, prediction, biomarker identification)

(d) Data Modality

(e) Number of Subjects

(f) Machine Learning Method(s)

(g) Validation Strategies

(h) Associated Outcome(s)

A full description of all performances according to each data modality can be found in S4 File.

## Study activities

To ensure all studies are categorised based on their different strategies and goals, each study was analysed based on study objectives into their identified activity:

(a) Diagnosis or detection of CI in PD (Comparison of data from PD patients with CI to PD patients with NC)

(b) Differential Diagnosis (Differentiating between PD with CI, and patients with other memory disorders)

(c) Condition progression prediction

(d) Identification of biomarkers for CI in PD

Each activity category can be linked to a type of ML technique needed to conduct the activities. Activities (a) and (b) focus on classification techniques, (c) focuses on prediction techniques, whilst (d) focuses on feature identification techniques.

## Study evaluation

Each study was scrutinised to identify its ML techniques, examining how they adapt to the challenge of detecting and diagnosing CI. Attention is given to how these techniques are adjusted to varying data types to determine those methodologies providing the most effective support across data types and activities. Whilst the impact on CI in PD is of increased importance, all studies use ML techniques, and therefore it is important to consider performances achieved by the different methods and their associated outcomes. Therefore, we compare achieved performance of ML techniques through analysis of their varying performance metrics. In studies using multiple ML models for analysis, the 'associated outcome' of the study is identified as the highest performing ML method(s) used. In studies encompassing training and validation phases, only validation performance was considered, and in the case that testing and validation are available, only testing performance is used. In studies performing multiple classification tasks, evaluation is centred around classification tasks focusing on distinguishing PD-NC from PD-CI/PD-MCI or PDD. Certain studies prioritise using ML techniques to draw specific conclusions, rather than concentrating on performance metrics. As a result, emphasis is placed on conclusions or findings obtained rather than numerical performance measures.

## Assessment of risk of bias

The risk of bias of all included studies was assessed based on the Prediction model Risk Of Bias ASsessment Tool (PROBAST) [112]. This tool examines 4 separate aspects of the study (participants, predictors, outcome, and analysis), with a number of signalling questions under each aspect marked as 'yes', 'no' or 'unclear' contributing to an overall assignment of risk of bias based on the study contents. Any assignment of 'no' indicates a high risk of bias and 'yes' considered a low risk. Overall risk of bias was considered low when all aspects are low and external validation was present, and considered high if any aspect was considered high, or all were low but no external validation was present.

## Observations and findings

### Literature review eligibility

Screening of all literature was performed in four stages. Based on the search criteria above, 1,052 publications were retrieved: 43 from PubMed, 252 from IEEE Xplore, 296 from Scopus and 461 from ScienceDirect. All duplicate publications were removed, excluding 32 results and all review papers were removed, excluding 299 results. 643 publications were removed based on title, abstracts and conclusions meeting exclusion criteria, and one publication was unable to be retrieved. 77 full-text publications were then screened for abstracts, methods, and conclusions. Seven further publications were excluded based on the exclusion criteria specified, resulting in 70 full-text articles available for analysis.

## Data sources

In 50 of the 70 studies, patient data was collected from recruited participants in one or more centres [21–31, 113–151]. 16 studies used data repositories, with 13 studies using data from the Parkinson's Progression Markers Initiative (PPMI) [152–165], and three studies using data from the National BioBank of Korea (NBBK) [84, 166–168], whilst four studies made use of data sourced from pre-existing research cohorts [169–172]. The average sample size was 184.72, with the smallest sample size of 17 [144] and the largest sample size of 2482 [131].

## Study activities

A number of study activities were utilised including diagnosis, (PD vs Healthy Controls (HC), PDD vs HC, PD-NC vs PD-MCI, PD-MCI vs PDD), differential diagnosis (PD-CI/PD-MCI/PDD vs AD vs DLB), identification of biomarkers for PD detection, and the prediction of future CI states. Most studies focused on diagnostic activities (n = 48) [21–26, 28, 30, 31, 84, 113, 114, 116–119, 121, 122, 124–126, 128, 130, 133–136, 138, 142, 144–146, 148–151, 153–155, 157–161, 164, 166, 171, 172], followed by prediction (n = 12) [29, 129, 131, 132, 137, 140, 141, 152, 156, 162, 163, 170], biomarker identification (n = 6) [27, 120, 123, 143, 147, 169], and differential diagnosis (n = 4) [115, 127, 139, 167].

## Data modalities

The most commonly used data modalities were imaging (n = 33) [24, 27–31, 113, 114, 117, 119, 122, 129, 130, 133, 137, 139–143, 147–152, 155, 156, 159–161, 163, 164], clinical characteristics (n = 17) [84, 118, 131, 132, 137, 140, 152, 153, 155, 156, 159, 160, 162–164, 169, 170], EEG (n = 11) [25, 26, 120, 125–128, 135, 136, 146, 151], and neuropsychological profile (n = 10) [115, 118, 132, 134, 142, 157, 158, 162, 167, 171], followed by a number of additional modalities. A clear overview of the population of discovered studies using each modality is found in Fig 2. A number of additional data modalities were only used in a singular study (n = 7) [116, 153, 162, 164, 170–172], with these being: eye movement, family history, environmental factors, Intelligence Quotient (IQ) & Emotional Intelligence Quotient (EIQ), biofluid assays, electronic health records, and smartphone test scores. Therefore, these remaining studies are grouped into a singular category of 'other'. A commonly identified theme in most reviewed studies is that the use of a singular data modality as a predictive feature is rare, but instead as part of a combination with other modalities. Therefore, discussions of data modality usage and outcomes focuses on all studies using a particular data modality, even when in combination with others.

## Machine learning techniques

ML techniques used across all reviewed studies were categorised into 12 categories, some of which overlap: (1) tree based methods (n = 32) [22, 23, 26, 27, 31, 114, 116, 119, 122, 125, 126, 128–130, 132, 134, 137, 139, 141, 145, 153, 155, 157–159, 162, 164, 166, 167, 170–172], (2) Support Vector Machines (n = 30) [21, 23–25, 27, 28, 30, 113, 115, 117, 118, 122–124, 133, 134, 138, 139, 141, 142, 148–151, 153, 156–159, 161, 172], (3) ensemble methods (n = 30) [22, 23, 26, 31, 114–116, 119, 122, 125, 126, 128–130, 132, 134, 139, 141, 145, 153, 155, 158, 159, 162, 164, 166, 167, 170–172], (4) regression based methods (n = 15) [30, 113, 115, 131, 134, 140, 152–154, 158, 162, 163, 167, 169, 172], (5) ANNs (n = 13) [29, 121, 122, 135, 136, 138, 139, 143, 147, 155, 157, 160, 162], (6) neighbour based methods (n = 12) [22, 23, 25, 27, 113, 115, 122, 127, 134, 155–157], (7) NB (n = 8) [23, 115, 133, 134, 139, 156, 157, 166], (8) DA (n = 4) [134, 146, 156, 166], (9) DR (n = 3) [21, 118, 120], (10) hybrid methods (n = 1) [84], (11) GP (n = 1)

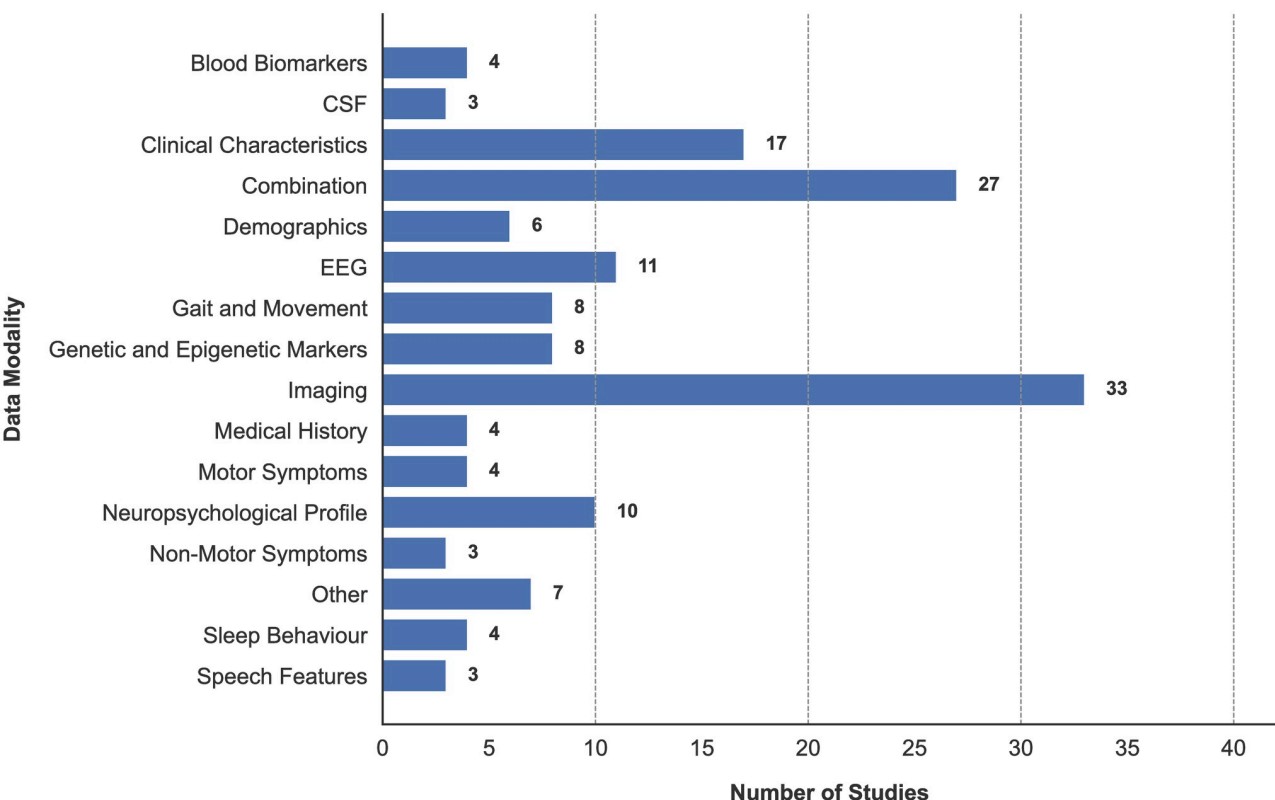

**Fig 2. Usage of data modalities across reviewed studies.**

[138], and (12) clustering (n = 1) [144], with most studies using at least two categories of ML model. A clear overview of the population of discovered studies using each ML technique is found in Fig 3.

As was stated above, all studies discussed in this review can be categorised based on the different study activity that they implemented, including either classification, prediction, or feature identification. Therefore, regarding usage of these study activities, and the ML techniques they use, the following sections discuss how each of these activities and ML learning types are employed across the studies that have been discovered, with considerations and descriptions given to the most notable studies. However, not all of these categories are populated, with some techniques not present for a particular learning type, and therefore discussions are made accordingly.

## Performance metrics

A considerable number of metrics have been used for the assessment of ML performance, as shown in Fig 4. The most commonly used metric was accuracy (n = 46), used both as a sole performance metric (n = 6) [25, 121, 123, 154, 155, 166] and as part of a combination with other metrics (n = 40). In studies using a combination of metrics, the most common combination was accuracy, sensitivity, and specificity (n = 17) [23, 113, 114, 116, 117, 124–126, 130, 133, 142, 149, 150, 158, 164, 167, 171], alongside accuracy, sensitivity, specificity, and Area under the ROC Curve (AUC) (n = 13) [22, 27, 31, 115, 119, 127, 135–137, 148, 151, 153, 161].

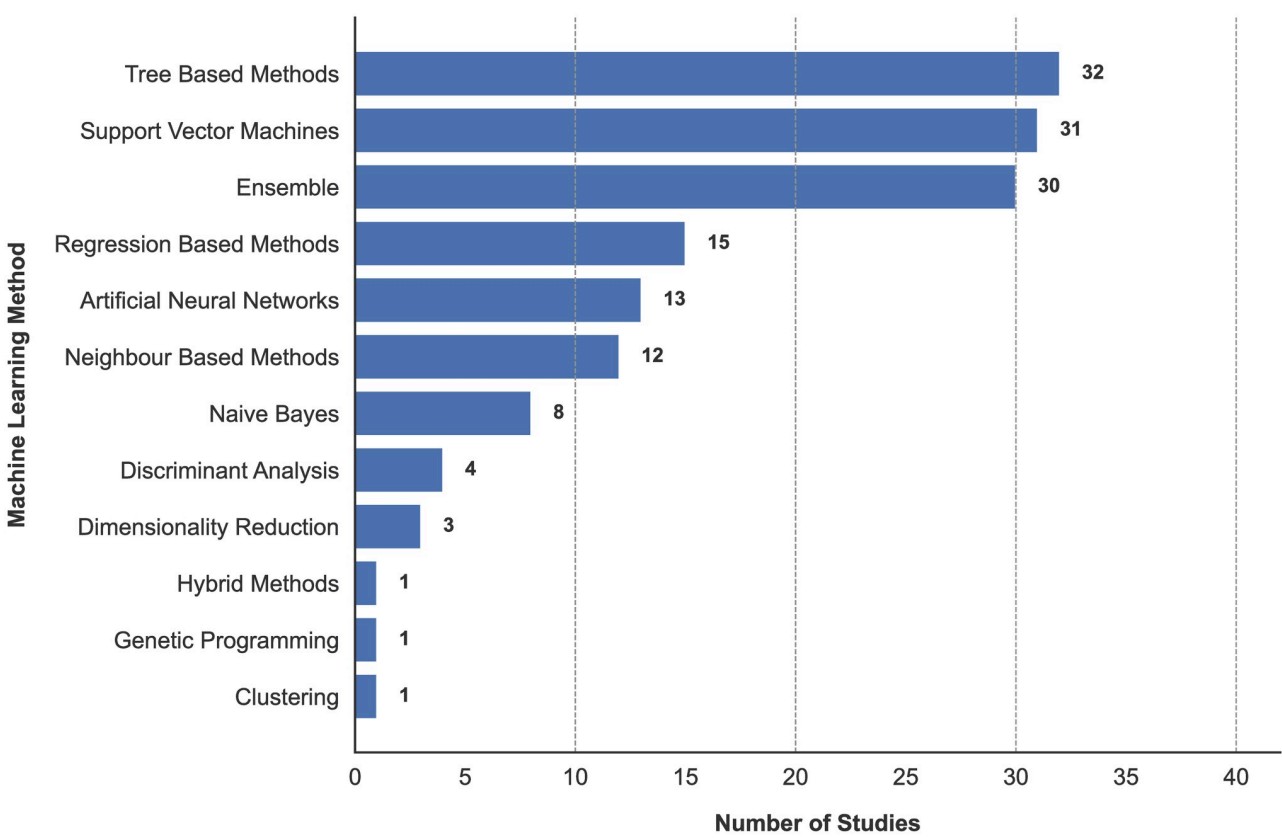

**Fig 3. Usage of ML techniques across reviewed studies.**

## Risk of bias assessment

An overall assessment of the risk of bias for all studies using PROBAST can be seen in Fig 5. For all included studies, 91.42% were deemed to be of a high risk of bias, and the remaining 9% having an overall low risk of bias. In the participant domain, 63 studies were judged to be at a low risk and seven judged to be at a high risk due to limited description of study data sources or inclusion and exclusion criteria. For the predictor domain, 69 studies were marked as low risk of bias and one marked as high risk. In the outcome domain, 60 were marked as low risk and 10 marked as high risk due to factors such as the inclusion of predictors in the assignment of outcomes. For the analysis domain, 52 studies were marked as low risk of bias and 18 marked as high risk largely due to an insufficient or largely imbalanced number of participants. Overall, a large proportion of studies were rated as high risk of bias, but for the majority of such studies, this assignment of high risk is due to the lack of external validation of the study.

## ML techniques applied in CI detection and diagnosis

All reviewed studies have employed supervised, unsupervised or deep learning techniques, with ensemble learning techniques falling under these banners. Overall, most studies discussed used supervised learning techniques in some form, with 58 studies utilising only non-DL supervised learning, two studies using unsupervised techniques alone, eight using DL techniques alone and two studies using a combination of supervised and unsupervised techniques.

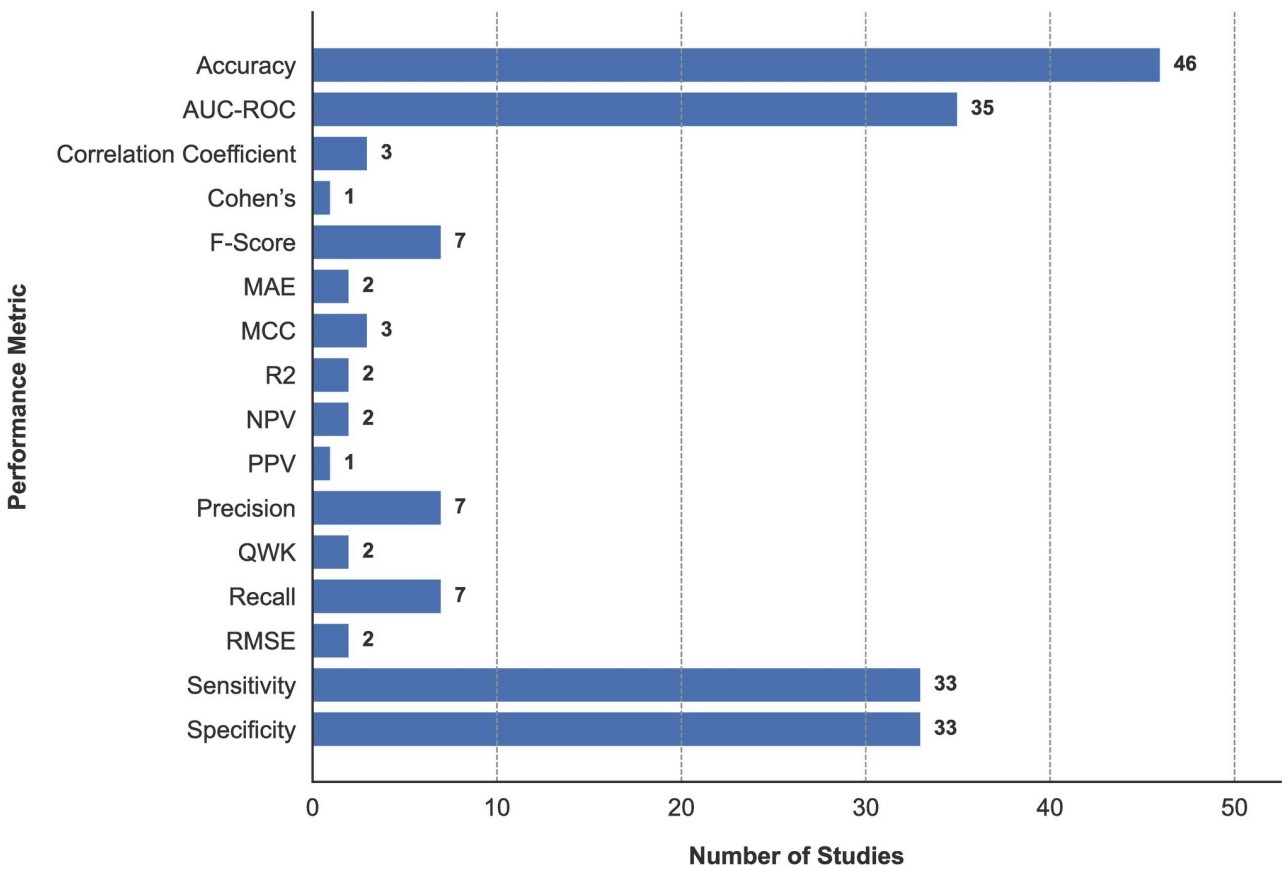

**Fig 4. Usage of performance metrics across reviewed studies.**

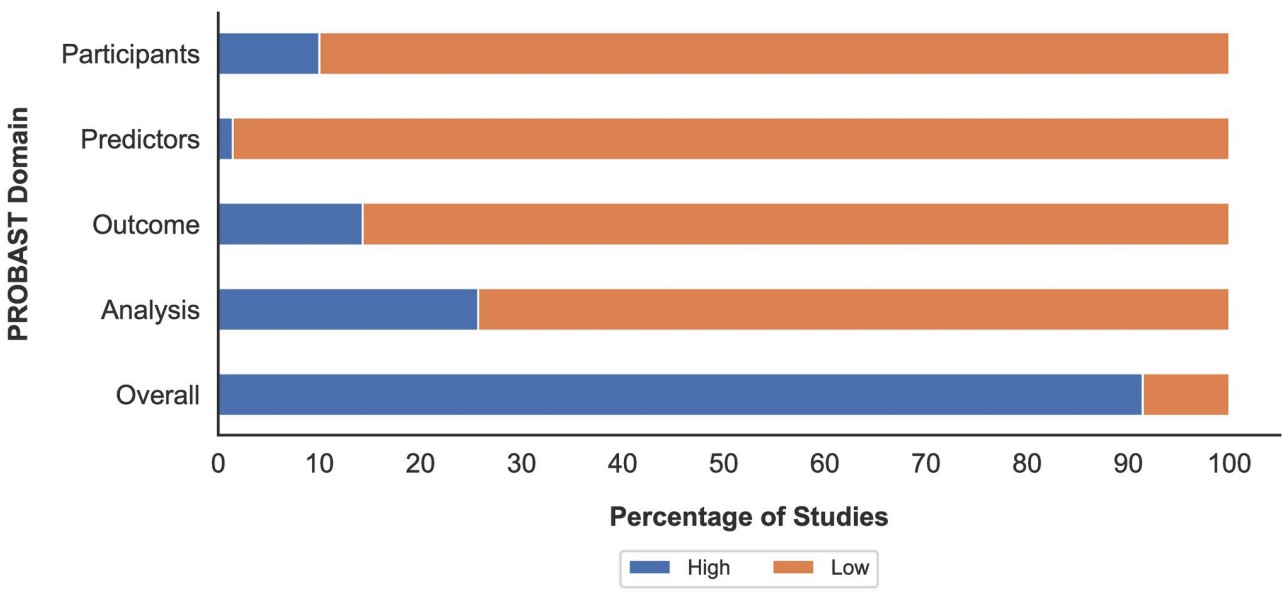

**Fig 5. Overall risk of bias assessment of studies using PROBAST.**

Since some studies used a combination of techniques from different categories for comparison, these studies are therefore grouped according to the supervision category held by the majority of methods or the most successful category. Therefore, those studies using combinations of supervised and unsupervised techniques are discussed under unsupervised learning. A complete description of all studies, the data modalities they used and their associated outcomes can be found in S3 File.

**Supervised learning.** 44 of the 58 supervised learning studies focused on classification activities, 40 of which focused on classification vs NC/HC [22–26, 28, 30, 31, 84, 113, 114, 116, 117, 119, 122, 124–126, 128, 130, 133, 134, 138, 142, 145, 146, 148–151, 153–155, 158, 159, 161, 164, 166, 171, 172], and four classifying against memory disorders [115, 127, 139, 167]. 11 studies focused on prediction [129, 131, 132, 137, 140, 141, 152, 156, 162, 163, 170], and the remaining three on biomarker identification [27, 123, 169]. Regarding modalities, 23 studies used a combination of modalities, with the remaining 35 using only one, with the most commonly used modalities being imaging (n = 29) [24, 27, 28, 30, 31, 113, 114, 117, 119, 122, 129, 130, 133, 137, 139–142, 148–152, 155, 156, 159, 161, 163, 164], clinical characteristics (n = 15) [84, 131, 132, 137, 140, 152, 153, 155, 156, 159, 162–164, 169, 170], and EEG (n = 8) [25, 26, 125–128, 146, 151]. The average sample size was 199.72, with the highest study using 2482 participants [131] and the lowest only 36 [146] As shown in Fig 3, tree based methods are the most commonly used technique in all studies, and this is evident in the supervised classification tasks conducted, with over 50% (n = 24) [22, 23, 26, 31, 114, 116, 119, 122, 125, 126, 128, 130, 134, 139, 145, 153, 155, 158, 159, 164, 166, 167, 171, 172] of studies using a tree model either alone or in combination. Notably however, the consensus with the method of choice is the use of an ensemble based tree method. RF is the most established classifier, used in 13 studies successfully. RF is shown to be a versatile model against a number of modalities, with RF models applied to 60% of the different modalities discovered. Byeon [166] compared performances of RF, NB, and DA models for the analysis of demographics, motor symptoms, non-motor symptoms, and sleep behaviour from 342 PD patients (66 Early Onset Parkinson Dementia (EOPD), 276 PD-NC). Overall, the RF model performed best, achieving an accuracy of 89.5%. Work by the same author [171] further demonstrated these benefits using a model trained on features from medical history, neuropsychological profile and environmental factors from 96 PD patients (45 PD-MCI, 51 PD-NC) compared to a singular DT on the same data. The RF model outperformed the singular tree, achieving an accuracy of 65.6%. Various modalities have emerged as commonly utilised sources for analysis of CI in PD. This trend is reflected in research using RF models. Koch et al. [128] analysed EEG data from 40 PD patients (20 PD-CI, 20 PD-NC) with an RF model, achieving an overall accuracy of 91% and an AUC of 0.98. In a different approach, Russo et al. [23] focused on analysing gait patterns to distinguish 40 PD-MCI and 40 PD-NC patients. They compared various ML models, including DT, RF, NB, SVM, and K-NN, with RF outperforming others with an accuracy of 81%. Lin et al. [130] examined connectivity features from MRI, DTI, and fMRI using an RF model with data from 179 subjects (59 PD-MCI, 72 PD-NC, 48 HC). Analysis yielded an accuracy of 85.2%, highlighting the effectiveness of RF models in handling various modalities to identify CI in PD.

RF models have also shown their abilities outside of the more commonly used modalities. Byeon [116] analysed IQ and EIQ data from 368 subjects (48 PD-MCI, 320 HC) using 9 tree based models combined of feature techniques (undersampling, oversampling, SMOTE) and ensemble methods (boosting, bagging, RF). The RF combined with SMOTE was capable of achieving the best performance, with an accuracy of 74%. Whilst this performance is not as high as values in other studies, the results show that there exists potential for CI detection

using features outside the scope of traditional modalities, and therefore should not be automatically excluded from consideration when developing models and clinical procedures.

Several studies have demonstrated the usefulness of other ensemble methods, particularly boosting in analyses of CI in PD. Chen et al. [119] examined DTI from 133 PD patients (52 PD-NC, 68 PD-MCI), comparing DT, XGBoost and RF models. The XGBoost model emerged as the top performer, achieving an accuracy of 91.67% and an AUC of 0.94. Tsiouris et al. [164] analysed an array of modalities, including medical history, clinical characteristics, imaging and more using DT combined with AdaBoost. Their analysis achieved a competitive accuracy of 80.38%. These findings suggest that boosting methods can exhibit similar performance levels to RF, emphasising the value of ensemble methods in general. However, the choice between models may depend on specific considerations, as boosting may not outperform in all scenarios.

A notable theme in discovered studies is the use of ML to analyse neuropsychological profiles alongside other modalities, particularly neurological assessments such as the MoCA. Jeon et al. [158] analysed MoCA values from 397 PD patients using RF, SVM and LoR models, achieving an accuracy of 0.88 with similar sensitivity and specificity values. These results showcase the potential for analysis of neuropsychological details using ML to enhance understanding of detection and diagnosis of CI normally done in person using clinician judgement.

Despite its prominence, DT and RF models are not the only models that have successfully been utilised for classification against NC and HC, with the K-NN, SVM, and NB showing considerable performance. Two studies in this review identified the K-NN as the most successful model in their respective analyses. Arslan et al. [113] while examining Arterial Spin Labelling MR imaging from 68 participants (26 PD-NC, 27 PD-MCI, 15 HC), found that the K-NN outperformed both the SVM and LoR models, achieving an accuracy of 92.60%. Similarly, Betrouni et al. [25] analysed resting state EEG from 118 participants in varying cognitive states, including cognitively intact, mental slowing, mild cognitive deficits and severe deficits. In their study, K-NN surpassed SVM, achieving an overall accuracy of 87%. These findings underscore the effectiveness of K-NN models in specific contexts when applied to diverse modalities.

Regarding NB, a sole study by Morales et al. [133] achieved successful classification of PD-MCI and PDD by exploring variants of the NB including traditional, multivariate filter and filter selective methods in comparison with SVM. Their analysis centred on MRI from 45 participants (16 PD-NC, 15 PD-MCI, 14 PDD). The filter selective method demonstrated the best overall performance when applied to MRI data, achieving an accuracy of 70% ± 26.66%. While NB models were less prevalent in the studies reviewed, this instance shows their potential in specific contexts for distinguishing between CI states in PD.

SVM has consistently demonstrated its ability in classification tasks, particularly with imaging data, as evidenced by numerous studies achieving AUC values up to 0.95. Abos et al. [24] employed SVM to differentiate between 133 subjects (60 PD-NC, 33 PD-MCI, 38 HC) using functional connectomics extracted from resting state fMRI. The SVM model achieved an overall AUC of 0.81, showcasing its efficacy in discerning cognitive states. Similarly, Kang et al. [30] achieved superior overall performance using an SVM to analyse magnetic susceptibility values and radiomics features from 149 subjects (22 PD-NC, 26 PD-MCI, 56 PDD, 45 HC). When compared with multivariate LoR trained on the same data, SVM demonstrated superior performance, achieving an AUC of 0.95. These findings underscore the robustness of SVM models in effectively handling diverse imaging modalities for the classification of CI in PD.

DA emerged as a less frequently employed technique, with only four instances of successful implementation. In one study by Tulay et al. [146], LDA was applied to EEG data collected during visual and auditory stimuli from 36 participants (Mild PDD, Moderate PDD, HC), aiming to differentiate between varying levels of PDD and HC, achieving an impressive

accuracy of 94%, with equally noteworthy values of 96% and 95% for precision and recall respectively. In another study by Ortelli et al. [134], QDA was used to analyse neuropsychological test scores from 500 participants, with the aim of differentiating between NC, MCI and impaired cognition. QDA outperformed various other models, including tree, regression, K-NN, SVM and ensemble, achieving the highest overall performance with an accuracy and AUC of 68.2% and 0.81 respectively. While DA may be less prevalent, these successful instances demonstrate its potential in specific contexts for discriminating among cognitive states in PD.

A noticeable trend in reviewed studies moves towards the adoption of unconventional methods for analysis. One such emergence is the use of GP and hybrid methods, which bring novel perspectives and innovative strategies to the study of CI. These methods represent an expansion of the analytical toolkit, offering researchers new ways to extract valuable insights from complex datasets and further enhance our understanding of these cognitive disorders. Picardi et al. [138] explored the use of Cartesian GP (CGP) alongside SVM and ANN to analyse gait and movement data from sensors on 85 subjects (22 PD-NC, 23 PD-MCI, 10 PDD, 30 HC). The analysis revealed that when comparing PD-NC to PD-MCI, SVM exhibited the best performance with an AUC of 0.78. However, in the comparison between PD-NC and PDD, CGP achieved the highest performance with an AUC of 0.83. Similarly, Byeon et al. [84] tackled the differentiation of PD-MCI from PD-NC using a hybrid approach. They analysed 48 diagnostic data variables encompassing motor and non-motor symptoms and sleep behaviour from 185 PD patients (75 PD-NC, 110 PD-MCI). Employing both hybrid and single models, with the hybrid models being combinations of Polydot, Vanilladot, RBFdot and C5.0, the study found that the combination of RBFdot and C5.0 hybrid models achieved the highest performance with an overall AUC of 0.88. These findings highlight the diverse range of ML techniques applied to the classification of CI in PD, demonstrating the potential for various methods to excel in specific contexts and tasks.

Of the four studies focused on differential classification, the most commonly successful ML method was the K-NN, successful in half of the studies, with the remaining two using RF and SVM. Two of these studies made use of a combination of data modalities as input features, whilst the remaining two studies used a singular modality. Jennings et al. [127] analysed resting state EEG data from 80 participants (32 AD, 26 DLB, 22 PDD), using a K-NN method and was able to differentiate between the different memory disorders with an accuracy of 61% ± 16%, and an AUC of 0.61. Byeon [167] also attempted to differentiate 110 PDD patients from 118 AD patients using a combination of sleep behaviour and neuropsychological profile using RF, LoR and classification and regression tree (CART) models. Study results found that the RF was most accurate in differentiating between the two patient groups with an overall accuracy, sensitivity and specificity of 73.3%, 78% and 70% respectively. Bougea et al. [115] differentiated between 78 probable PDD and 62 probable DLB patients using binomial regression (BR), K-NN, SVM, NB and an ensemble classifier of all models using features extracted from demographics and neuropsychological profile. Overall, the K-NN model achieved the highest overall performance, with an AUC of 0.958 and an accuracy of 91.2%. Rallabandi et al. [139] attempted to differentiate between 27 PDD, 30 MCI, 27 AD and 33 HC using DT, RF, NB, MLPs and SVMs using features from T1 Weighted MRI. Overall, the SVM model achieved the best performance, being able to differentiate between the different disorders with an AUC of 0.892. Of the 11 studies that made use of supervised ML to perform predictive activities, eight studies focused on the prediction of future cognitive assessment scores [129, 131, 132, 140, 152, 162, 163, 170], whilst the remaining three studies focused on the prediction of conversion from PD-NC to MCI and PDD [137, 141, 156]. In terms of modalities, imaging was the most commonly utilised modality, used in seven studies both alone and with other modalities [129,

137, 140, 141, 152, 156, 163], whilst the remaining four studies used differing combinations of other metrics including clinical characteristics, genetic and epigenetic data, blood biomarkers, CSF, and neuropsychological profile among others [131, 132, 162, 170]. Overall, tree based models were the most commonly successful ML method used in prediction activities, being used in six studies [129, 132, 137, 141, 162, 170], followed by regression in five studies [131, 140, 152, 162, 163]. In studies using tree models, the most commonly successful model was the ensemble RF model, which was used as a single predictive model in three of the six studies [129, 132, 170], and two studies comparing the performance of the RF to other models [141, 162], with a singular study using the extra trees classifier [137].

Kubler et al. [129] analysed VBM features of the Nucleus Basalis of Meynert extracted from MRI of 55 PD patients (19 PD-NC, 36 PD-MCI) and was able to predict the cognitive outcome of PD patients with an average RMSE of 11.28 ± 9.51%. Lo et al. [170] analysed scores gathered from smartphone tests alongside clinical characteristics to predict cognitive outcomes of 237 PD patients from the Oxford Discovery Cohort with an overall AUC between 0.75 and 0.97. Shin et al. [141] analysed T1 Weighted MRI data collected from 141 PD patients, 46 who converted to PDD and 95 who did not to discover important clinical and cortical thickness features. This imaging data was used to train both RF and SVM (Linear, Polynomial, Radial, and Sigmoid Kernels) models, with the RF model achieving an AUC performance of 0.84, indicating the benefit of cortical thickness in conversion from PD-MCI to PDD. McFall et al. [132] used a combination of features including demographics, gait, blood biomarkers, genetic data, clinical characteristics and neuropsychological profile to predict incipient dementia after three years from non-demented PD. Data was gathered from 48 PD patients (34 PD-ND, 14 PD-ID), and identification was able to be made with an overall AUC of 0.85 and an accuracy of 81%.

In studies where RF was not the most successful, Salmanpour et al. [162] compared a number of alternative tree models and other methods including Local Linear Model Trees, Radial Basis Functions, Multilayer Perceptrons, and Thiel-Sen regression for predicting the MoCA score at year four based on the previous four years of longitudinal data gathered from 492 PD patients. Overall, the Local Linear Model Trees model was found to have the superior performance in predicting the MoCA score at year four when combined with a NSGAII genetic sorting algorithm, achieving an overall MAE score of 1.68 ± 0.12. Park et al. [137] used the RF model to predict the conversion of PD-NC to PDD using a combined model of imaging and clinical characteristics from 262 patients (75 PDD, 187 No-PDD) and was able to predict conversion to PDD with an overall accuracy of 79.8% and an AUC of 0.89.

Three separate types of regression were used for predictive activities. Liu et al. [131] used Cox regression to create an overall cognitive risk score for the prediction of future cognitive state within 10 years of disease onset based on a dataset of features combining clinical characteristics and genetic and epigenetic data. CI was predicted within 10 years with an AUC of 0.85, whilst PDD was predicted with an AUC of 0.88. Ramezani et al. [140] utilised a regression adapted SVM combined with RReliefF feature selection to investigate the ability to predict future cognitive states based on SNCA gene status. 101 PD patients were used, from whom imaging, genetic, epigenetic, clinical characteristics and demographics were extracted and used to train the feature extraction and regression. 11 features were found to be predictive of global cognitive decline in PD, with a clear association given between the rs894280 SNCA gene and global cognition, validated by a correlation coefficient of 0.54. Schrag et al. [163] used a LoR model trained on clinical characteristics, genetic and epigenetic data, CSF and DAT imaging gathered from 568 subjects, 390 with PD and 178 healthy controls. This model was then used to predict the overall change in MoCA score from a baseline to two years post baseline. Five variables were found to show the most significant association with CI and allowed for

prediction of cognitive decline at the two year stage with an AUC of 0.80. The most commonly used ML technique was the SVM model, used twice [27, 123], whilst tree, neighbour and regression were all used once [27, 123, 169]. Deng et al. [169] used a combination of clinical features, blood biomarkers and genetic and epigenetic data from 206 participants (108 PD-MCI, 98 PD-NC) in the Early Parkinson's Disease Longitudinal Singapore Cohort. Shape-lyVIC [173] and Backward Selection feature selection techniques were used to determine the most likely variables associated with PD-MCI. A subset of 22 variables were identified as significantly important to the development of PD-MCI and were then used as training parameters for a multi-variable log-binomial regression to determine the relative risk of each variable to the development of MCI in PD, in which a selection of the eight most common variables were made, including education years, hypertension history, higher levels of triglyceride and apolipoprotein, and higher MDS-UPDRS scores.

Both SVM studies were the only studies to utilise performance metrics as an overall outcome with regard to scientific conclusions, whilst the remaining three studies focused on making scientific conclusions and did not include any performance metrics. Of the two studies, one used SVM as a singular identification model [123], whilst the remaining compared the performance of a SVM, K-NN and tree based models (J48 DT, adaBoost, RF) [27].

Work by Garcia et al. [123] analysed speech features from 80 participants (16 PD-MCI, 24 PD-NC, 40 HC). Prosodic, articulatory, and phonemic identifiability speech features were extracted from both groups during reading and retelling tasks. These features were then used in training SVM models with a Gaussian kernel to determine the ability for differentiation between each patient group using each feature type. Overall, it was discovered that in regard to the differentiation of PD-MCI from PD-NC, the use of phonemic identifiability as the main feature functions as the best approach for use as a speech biomarker for CI, with an overall accuracy of 72.1% when utilised during the retelling task, and accuracy of 71.9% during the reading tasks. Similar results were found when differentiating PD-MCI patients from HC, with overall classification achieving accuracies of around 86.9% during the retelling task. Both of these results clearly show the viability for the phonemic identifiability feature as a speech based biomarker for the identification of PD-MCI patients from cognitively preserved patients and healthy controls simultaneously.

In the comparison study conducted by Amboni et al. [27], the SVM model was shown to achieve higher performance than the K-NN and tree based models. This study focused on the dual analysis of Amyloid PET imaging and gait data gathered from 75 PD patients, 33 with MCI and 42 without. Features from these two modalities were used to create three training data variants: variant one employing clinical, spatial and temporal gait variables, variant 2A containing the top five features from variant one and averaged amyloid PET retention from all brain regions, and variant 2B employing the top five features and average PET retention from only cortical areas.

These three variants were then used to train all model comparison types. The SVM model trained on variant one achieved the highest overall performance with an accuracy and AUC of 80% and 0.792 respectively. The high performance of this model therefore indicates that gait features can function as a more superior biomarker than Amyloid PET imaging for the development of CI in PD.

**Unsupervised learning.** Four studies used unsupervised learning techniques to analyse CI in PD, two used unsupervised alone [120, 144] and two used unsupervised in combination with supervised techniques [21, 118]. Of these studies, three focused on classification (PD-MCI vs PD-NC) [21, 118, 144] and one focused on biomarker identification [120]. Regarding modalities, EEG [120], gait [21], clinical characteristics, blood biomarkers, neuropsychological profile [118], and speech features [144] were used once. The average sample size

was 43.25, with the highest study using 81 participants [21] and the lowest only 17 [144]. Two studies [21, 118] conducted a combination of supervised and unsupervised techniques using an SVM combined with PCA (PCA-SVM) to classify PD-MCI from PD-NC. These studies used gait and movement data, and a combination of clinical characteristics, blood biomarkers, and neuropsychological profile to achieve (accuracy, AUC) scores of (91.67%, 0.9714) [21] and (92.3%, 0.929) [118]. This clear performance benefit compared to SVM methods that achieve accuracies and AUC values in the range of 70–80% indicates that analysis of CI can benefit from reducing the high dimensionality of data to a smaller set of components. Talkar et al. [144] leveraged the benefits of clustering to classify PD-MCI from PD-NC. Speech and motor coordination features from nine PD-NC and eight PD-MCI patients during reading tasks were used to train a GMM that was able to accurately classify with an AUC of 0.84. This high performance indicates that classification of CI in PD may be suited in some cases to the use of clustering alongside other ML techniques, however this will require further validation across other modalities. Chu et al. [120] used DR based feature identification to analyse EEG data from 33 PD patients (13 PD-MCI, 20 PD-NC) to discover biomarkers for MCI development. This EEG trained an NMF model based on sliding window techniques to track sub-networks in functional brain networks. From this, the authors identified a biomarker indicating that five functional sub-networks function as part of a network of early PD patients, with MCI inducing slow, interrupted evolution of these.

**Deep learning.**   13 studies used a DL model in some form, either in comparison or alone. This section focuses solely on the eight studies where it was the most successful model. Five of these studies focused on classification [121, 135, 136, 157, 160], two focused on biomarker identification [143, 147] and one focused on prediction [29]. Most studies used a single modality, with only two studies using a combination. The most commonly used modalities were imaging (n = 4) [29, 143, 147, 160] and EEG (n = 2) [135, 136]. The average sample size was 146.125, with the highest study using 476 participants [157], and the lowest using only 27 [143]. A common theme in classification studies is the comparison of proposed models to determine superior performance. However, in DL studies, this consensus does not remain, with most studies only analysing the overall performance of the model (n = 4) [121, 135, 136, 160], with a single study analysing the model in comparison [157]. Ismail et al. [157] attempted to identify PD-MCI and PDD in the neuropsychological profile of 467 patients from the PPMI database using features including MoCA, Semantic Fluency, and patient age. Features from this profile were used to train six classification models: DT, SVM, K-NN, NB, RF and a custom four layer MLP. Out of all six models, the MLP achieved the best performance, with an accuracy and AUC of 97.5% and 0.995 respectively.

Image analysis and classification are tasks in which DL techniques are most utilised since they adapt well to a number of different image types and content, even in the same dataset. This common usage lines up to the large proportion of studies using imaging as a modality, either alone or in comparison. A common principle in image based DL work is transfer learning, in which model parameters from a similar, successful problem are adapted to a new problem as a basis for training.

Ostertag et al. [160] used this technique to analyse MRI combined with clinical characteristics from 134 PD patients (47 stable cognition, 87 declining cognition) with a dual analysis DL model in which each modality is given its own independent analysis architecture before making a joint decision. This network was pre-trained on learned data from a model used in detecting AD in ADNI before being trained on PPMI data. Transfer learning had a considerable impact on performance, increasing the AUC to 0.81, compared with 0.72 on the PPMI data alone.

DL techniques are also effectively applied to other data modalities and formats, including structural and tabulated data. This versatility is exemplified by studies such as Ismail et al. [157] who leveraged neuropsychological assessment values, and Chung et al. [121] who examined levels of plasma-borne circulatory tau, $\beta$-amyloid, and $\alpha$-synuclein carried by extracellular vesicles, age and sex as predictive factors. Chung et al. [121] analysed levels of tau, $\beta$-amyloid, and $\alpha$-synuclein in the blood stream, age and sex in training a 3 layer MLP. Their model achieved an accuracy of 91.3% and an AUC of 0.911 during validation. Moreover, these findings shed light on the association between plasma EV tau and cognitive functions, with elevated levels of EV tau and A$\beta$1–42 in CI patients. These discoveries highlight the potential of DL in understanding cognitive dysfunction, emphasising its broader applications beyond imaging. In line with the above theme regarding transfer learning, work by Choi et al. [29] implemented this technique through the use of an identified cognitive signature of FDG-PET imaging of the brain in patients with AD. This cognitive signature was identified with a custom CNN built for classifying AD patients, and was then used as part of the transfer learning procedure to train a separate CNN model for predicting PD patients with MCI who would convert to PDD. When transferred to this model for identifying conversion from MCI to PDD, the overall model was able to predict with an AUC performance of 0.81. This clearly shows the potential benefits for the training of DL models on a framework of previous data and knowledge extracted from tasks trained on different memory disorders. A strong benefit of using DL models for image analysis is the ability to visualise and interpret the learning process using activation mapping (AM) [174], which enable understanding which parts of an input image are most influential in prediction and are used the most by a model during decision making. Therefore, the usage of such techniques can be used to identify brain regions influential in CI in PD. Suwalska et al [143] used AM in a custom CNN to assess T1 and T2 weighted MR sequences to identify the most prevalent brain areas during training. MRI data was gathered from 18 PD patients (10 PD-NC, 4 PD-MCI, 4 PDD) and used to train the CNN model to differentiate each patient group. AMs were constructed per patient to identify the regions used most in the classification process, and areas with the highest average across all patients considered the most notable. Overall, this work found the severity of CI can be assessed on regions identified during model training, with the cerebellum being the most significant in differentiating patient groups. On a similar theme, analysis of DL models can be used to determine the categories within data that are the most influential for a decision, and can lend themselves well to identify the most crucial criteria in data. Xu et al. [147] used such techniques with a CNN to analyse Regions of Interest in Hippocampal Mapping Images (HMIs) of T1 MRI gathered from 245 PD patients (195 PD, 25 PD-NC, 25 PD-MCI) alongside HC, MCI and AD patients. Subfields in these HMIs were used to classify these patient groups. Performance results identified a number of areas that most accurately classified patient groups and function as important biomarkers for MCI in PD including the left parasubiculum, left HATA, and left presubiculum.

## Discussions

The aim of this systematic review was to evaluate the effectiveness of ML techniques in the detection and diagnosis of cognitive impairment in PD. Our comprehensive analysis of the selected studies has highlighted the significant potential that these techniques have to enhance diagnostic accuracy and provide alternative, non-invasive methods to traditional assessment methods.

Our key findings have demonstrated that ML is capable of analysing various data modalities—such as imaging, speech patterns, gait and EEG—to effectively identify the presence of CI

in PD, showing a capability of ML to handle a wide array of data types. Notably, the application of ensemble learning models, particular the RF model, has shown promising results across multiple studies, indicating a robustness for handling diverse data.

Moreover, ML methods have been shown to have potential in identifying early markers for CI development, often before they clinically manifest, which can be crucial in preparing timely interventions to impact disease progression. Several studies have also recorded significant diagnostic accuracies, suggesting that ML could significantly improve precision of diagnostic processes.

## Data collection

As is the case in the broader medical research landscape, studies examined in this review prioritise data from human participants. This preference stems from the inherent relevance and authenticity of data obtained directly from those with the condition under investigation. Engaging with participants directly allows researchers to gather comprehensive insights into the target population, thereby enhancing overall quality and depth of findings. Whilst other data sources have been used and are valuable in some contexts, emphasis on first-hand collection underscores the fundamental importance of real-world patient perspectives in advancing understanding and treatment of medical conditions. However, it is worth noting that some studies rely on data from a single geographical restricted centre, potentially limiting generalisability of findings. To address this and enhance results reliability, further validation efforts are warranted. This could involve study replication in diverse geographical locations or leveraging online databases such as the PPMI to validate and extend findings.

A positive trend worth highlighting is the increasing use of online data sources to access large-scale data required for training ML models, particularly ANNs. This shift is likely driven by recognition that online databases offer extensive and diverse data opportunities crucial for training accurate and generalisable ML models. Importantly, databases such as these facilitate continual growth, ensuring that ML techniques can evolve and improve as more data is available. This ongoing expansion of resources holds promise for developing more robust and applicable ML models in clinical research.

Commonly utilised methods have involved use of bodily data such as imaging and EEG, however it is essential to recognise the significance of considering CI from a variety of perspectives and modalities to encompass all factors affecting the condition. This is significant and apparent when considering the substantial proportion of studies using more than one modality in analysis. Use of a multifaceted approach underscores the need to encompass a comprehensive range of data sources to provide a holistic understanding of the subject matter. Furthermore, clinical characteristics have shown to play a pivotal role in the research landscape, as evident from their inclusion in 17 discovered studies. This demonstrates the prevailing importance of clinical data as a feature in predictive models. These characteristics provide critical insights into development and progression of CI, further emphasising this need for a well-rounded approach in the pursuit of robust and accurate predictions.

## Model selection

As shown in Fig 3, employment of ML methods predominantly revolves around tree, SVM and regression techniques. This is likely attributed to the use of a diverse array of individual and varying features. These features encompass a broad spectrum of data modalities, aligning well with the capabilities of these techniques. The inherent adaptability of these techniques make them particularly suited for the handling of the complex and varying nature of these features, contributing to their prominence in analysing CI across diverse data types. Ensemble

methods, particularly the tree based RF, stand out as a prevalent technique. These methods have gained popularity due to their ability to combine various predictors of different types and strategies, creating a unified decision-making process. This amalgamation of predictors serves the crucial purpose of simplifying complex data, which is increasingly valuable when dealing with intricate and multi-faceted data modalities, something that is prevalent in CI research.

The adoption of DL techniques, particularly CNNs tailored for specific analysis tasks has also been notable. This surge is likely driven by the inherent advantages of DL techniques, which excel at handling complex datasets such as medical imaging data. DL models possess the unique capability to autonomously learn meaningful data representations without direct instruction and are increasingly adaptable; these architectures can be designed and fine-tuned precisely for a particular task, rather than relying on standardised models requiring data adaption. As a result, these techniques have shown considerable performance in the identified studies, particularly those that are image based, with performance accuracies as high as 80–90%+.

However, it is important that traditional models are not disregarded in tasks like these, with conventional ML approaches continuing to demonstrate impressive performances, achieving accuracies ranging from 70% to 95% depending on data modality employed. As such, the choice of model should always be guided by dataset characteristics. Less complex datasets have yielded excellent results with traditional models, whilst more intricate data, such as neuroimaging data benefit from the sophistication and adaptability of DL models. Whilst the appeal of more advanced models is undeniable, it is still important to select the most appropriate model for the unique attributes of a dataset. The decision should be driven by data complexity and nature, ensuring that the chosen model aligns optimally with research goals and analytical requirements.

## Performance analysis

A majority of studies opted to employ accuracy as a primary performance metric. Within these studies, diagnostic levels were consistently elevated, surpassing natural chance substantially. Notably, accuracy levels spanned a broad spectrum, from an impressive 99.72% to a noteworthy 60.5%. A notable trend is that those studies employing CSF data chose to utilise other performance metrics, including AUC, MAE, R2 and RMSE. This diversity in approaches underscore the inherent flexibility of ML in catering to specific demands of CI analysis scenarios, whilst identifying the prominence of accuracy as a pivotal benchmark across a range of techniques and modalities.

## Condition classification and prediction

This review has demonstrated the substantial benefit of employing ML techniques for classification and prediction of CI in PD. These investigations have identified the efficacy of supervised, unsupervised and deep ML approaches to achieve crucial successes. Supervised techniques involving training models on labelled data of known outcomes have showcased their ability to accurately classify CI in PD patients. These methods evidently leverage the wealth of data available to discern patterns and relationships that may be challenging in traditional analyses. Learning from historical and longitudinal data allowed supervised ML models to make informed predictions about an individuals CI status, thereby aiding in early detection and intervention.

Similarly, unsupervised ML techniques, which excel at identifying hidden patterns and structures in data have proven their worth for this domain, contributing to a deeper understanding of CI in PD. These techniques have shown themselves to be useful in identifying subtle groupings or associations between patient groups, identifying potential risk factors or

disease sub-types that might have gone unnoticed in traditional analyses. Collectively, these studies have underscored the utility of ML techniques in enhancing the ability to classify and predict CI. ML has shown itself as a powerful tool for unravelling condition complexities, offering potentially more accurate diagnoses and personalised treatment strategies in the future.

The adoption of transfer learning stands out as a significant element in this realm, offering numerous benefits, with the chief among them being the ability to establish a foundation of knowledge expanding outside the domains of PD, MCI and PDD. PD-MCI and PDD share symptoms, presentations and biological underpinnings with other memory disorders and CIs. By applying transfer learning, researchers have shown the possibilities for using this wealth of insights and representations garnered from these related domains. Therefore, instead of relying on ML systems to learn entirely new representations for tasks involving PD-MCI and PDD, researchers can build upon this existing knowledge. Using models pre-trained on AD, researchers have benefited from the insights and representations developed in that domain [29, 160]. This approach acknowledges that PD-related cognitive impairments, such as PD-MCI and PDD, often exhibit overlapping symptoms and underlying mechanisms with Alzheimer's Disease. Instead of starting from scratch, transfer learning allows for a smoother transition in addressing the cognitive aspects of PD by building upon the established foundations of Alzheimer's research. Transfer learning not only facilitates transition but also enables fine-tuning of existing models to better address specific symptoms and progression patterns of PD and CI. This process benefits the model itself by enhancing its overall accuracy and adaptability and enabling it to generalise more effectively to new, unseen data whilst maintaining robust predictions against diverse patient profiles which as a result causes the model to become more versatile and reliable for clinical applications.

This approach saves time and resources, but also enhances ML model robustness and efficiency by enabling the development of more accurate predictive models and better informed decision making processes. The knowledge base established with transfer learning can also facilitate cross-disciplinary collaborations and the adaption of insights across domains, accelerating progress in understanding and management of CI. It also highlights the importance of previously acquired knowledge as a valuable resource for further study and encouraging research to consider outside the boundaries of individual conditions.

## Biomarker identification

Feature selection methods underscore the need for a diverse array of data sources beyond data collected through physical examinations. Whilst this data holds clear importance, it must be recognised that they should not be an exclusive option for insights in medical analysis. Therefore, it is unsurprising that many studies reviewed make use of a broad spectrum of data modalities when attempting to discover biomarkers for CI. In PD research, a wealth of evidence has accumulated showing the promise of a variety of features in predicting the onset and progression of CI. These features encompass diverse data modalities, reflecting the complex, multifaceted nature of the disease. These findings underscore the importance of considering a wide array of variables and sources for understanding CI.

Numerous studies have affirmed the potential for identifying biomarkers of CI development in PD. Studies in this review have highlighted the versatility of multiple models for identifying these biomarkers, indicating the need for continued research. The utilisation of methods such as the SVM and ANN have garnered validation, underscoring their promise in pinpointing biomarkers across a range of modalities alongside other models. The results of these studies each provide empirical support for the capability of SVM and ANN models to

discern valuable biomarkers from various modalities. This further underscores the robustness and adaptability of these computational approaches in handling the intricate task of biomarker identification. Feature selection methods have been widely employed in the context of CI in PD, revealing a substantial array of distinct features that hold potential for identifying the development of CI in PD patients. This underscores the valuable role of ML techniques in uncovering biomarkers for CI progression.

In the pursuit of understanding and predicting CI within the PD population, feature selection methods have emerged as essential tools. These methods have systematically combed through a diverse set of data attributes, going beyond solely relying on bodily collected features, such as those obtained from medical examinations like imaging and EEG. While these clinical assessments are undeniably important, feature selection methods have illuminated the fact that they should not be viewed in isolation. Instead, they should be complemented by a broader spectrum of data sources. By leveraging ML and feature selection techniques, researchers have identified a multitude of features that can be harnessed for the early detection and monitoring of CI in individuals with PD. These features encompass a wide range of data types, including genetic information, demographic factors, lifestyle choices, and environmental variables. The clear benefit of incorporating this diverse set of features is that it enhances the accuracy and robustness of predictive models, allowing for more precise identification of CI risk factors and progression markers.

Furthermore, the application of ML in biomarker discovery for CI in PD has demonstrated its potential to revolutionise the field of medical research and clinical practice. ML models, armed with the insights from feature selection, can aid healthcare providers in offering personalised treatment plans and interventions to PD patients at risk of developing CI. This holistic approach not only improves patient care but also paves the way for more effective strategies in managing and mitigating CI, ultimately enhancing the quality of life for individuals living with PD.

## Research challenges, limitations and recommendations

A notable concern in multiple studies is the use of small sample sizes (<50 subjects), with the lowest subject count being 17 [144]. Whilst the advantages of applying ML to various outcomes are evident, these smaller sample sizes can present challenges for models like ANNs that typically demand datasets with tens of thousands or more samples to achieve optimal performance. Gathering such extensive data is intricate, especially in clinical settings focused on niche areas. Therefore, alternative approaches that ensure the usability of limited data are crucial.

Similarly, since many studies originate from a clinical rather than computational perspective, a noticeable gap emerges concerning specifics regarding employed ML techniques. Key details including architectures, parameters, programming language, training, testing and evaluation strategies are often absent. This absence of information renders the replication of such research a challenge. To ensure findings are practical and reproducible, it would be advantageous for researchers to incorporate these elements into their studies. This inclusion would facilitate a higher degree of replication and implementation, fostering more robust advancements in ML for CI analysis. Additionally, a recurring theme throughout the studies in this review is the absence of a consensus regarding the categorisation of patients into different cognitive stages. Some studies adopt categories like NC, MCI, and PDD, while others employ terms including mild PDD, moderate PDD, impaired cognition, cognitively intact, mental slowing, mild cognitive deficits and severe deficits. This lack of uniformity in terminology used to define distinct cognitive states presents a challenge when attempting to draw clear

parallels and commonalities between various methods and modalities for identifying CI in PD, potentially complicating efforts to synthesise findings and establish standardised assessment criteria.

A recurrent theme in the studies identified is the absence of comprehensive descriptions regarding validation of approaches in a clinical context. These studies emphasise applying ML to CI, but the omission of detailed information is apparent, raising concerns about the practical applicability and reliability of these approaches beyond research settings. Ensuring the clinical robustness and effectiveness of these approaches becomes essential, and studies should incorporate validation protocols that encompass intricacies of real clinical scenarios.

Our assessment using the PROBAST tool has also revealed a concerning prevalence of studies with potentially high risks of bias in the majority of included studies. Specifically, the lack of any external validation stands out as a critical shortfall of the approaches, indicating a significant challenge that still exists with regard to the adoption of ML technologies and the need for comprehensive external clinical validation. Such issues are further increased by issues with descriptions of data sources, the mis-allocation of predictors and outcomes and increasingly small or imbalanced study populations which has the potential to skew overall study outcomes. Ensuring a balanced and comprehensive number of participants is crucial to mitigating any domain biases and ensuring the conduction of external validation can further enhance the credibility and applicability of such research.

## Conclusions

In answering the research question posed, this review has provided a comprehensive overview of studies applying the use of ML techniques to the detection and diagnosis of CI in PD. In this work, we have discussed the included studies in detail, covering data sources and modalities, sample sizes, ML methods and their associated outcomes, as well as identifying the most successful data modalities for use in ML based clinical analyses, potential biomarkers that could be used in clinical processes, and the current availability of large scale online databases for analysis of CI in PD. We have shown consistently that ML has potential for inclusion in diagnostic processes for detecting and diagnosing CI in PD. An inherent limitation of this study lies in its exclusive reliance on research published in English. This approach potentially overlooks valuable contributions from non-English sources that could have significant impact. By confining analyses to English-language publications, this work may inadvertently exclude valuable insights, methodologies, and findings from alternative contexts. Another constraint stems from the variation in information provided by studies, with pertinent details necessary for comprehensive comparisons lacking. The absence of key elements, such as descriptions of data splitting methodologies, training parameters and cross-validation strategies pose challenges in assessing the robustness and generalisability of the findings. As a result, the scope of this work is primarily confined to delivering a broad and general overview of research outcomes that centre around distinct modalities and ML methods. However, as shown, some studies are still limited and may require more investigation, largely to limit potential biases and issues arising from the use of small scale participant numbers, something that is not sufficient for use in most ML models. Analysis employing the PROBAST tool identified a concerning prevalence of high risk of bias, with 91.42% of studies classified as high risk due largely in part to a lack of external validation efforts. This indicates potential for significant challenges in the current research area, and emphasises the need for methodological improvements and adherence to validation practices in ML studies utilising clincal data. Despite these challenges causing an increase in overall risk of bias, the identification of a large proportion of studies with low risk of bias across multiple domains demonstrates the feasibility of conducting such

research and it is imperative to address these biases to ensure the applicability and integrity of any further research efforts.

## Supporting information

**S1 File. Search strings.** Search strings used for retrieval of publications from databases.
(PDF)

**S2 File. Secondary search keywords.** Secondary search keywords used for retrieval of publications from databases.
(PDF)

**S3 File. Summary of characteristics of reviewed literature.** Referenced Studies, Publication Year, Data Sources, Research Activities, Modalities Employed, Subject Details, Machine Learning Methods, Validation Techniques, and Reported Outcomes.
(PDF)

**S4 File. Summary of performance per modality.** Performance and characteristics of ML methods used for each data modality in reviewed literature.
(PDF)

**S5 File. PRISMA.** PRISMA checklist.
(PDF)

## Author Contributions

**Conceptualization:** Callum Altham, Huaizhong Zhang, Ella Pereira.

**Formal analysis:** Callum Altham.

**Investigation:** Callum Altham.

**Methodology:** Callum Altham, Huaizhong Zhang, Ella Pereira.

**Project administration:** Callum Altham.

**Supervision:** Huaizhong Zhang, Ella Pereira.

**Validation:** Huaizhong Zhang, Ella Pereira.

**Visualization:** Callum Altham.

**Writing – original draft:** Callum Altham.

**Writing – review & editing:** Huaizhong Zhang, Ella Pereira.

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
