## [Decision Letter · Decision Letter 0]

9 Apr 2024

PONE-D-24-09980Machine learning for the detection and diagnosis of cognitive impairment in Parkinson’s disease: a systematic reviewPLOS ONE

Dear Dr. Altham,

Thank you for submitting your manuscript to PLOS ONE. After careful consideration, we feel that it has merit but does not fully meet PLOS ONE’s publication criteria as it currently stands. Therefore, we invite you to submit a revised version of the manuscript that addresses the points raised during the review process. The reviewers have reviewed your manuscript thoroughly. While both reviewers have found your paper of great importance, they have both provided detailed and valuable feedback that can improve the quality of the paper. 

We look forward to receiving your revised manuscript.

Kind regards,

Farzin Hajebrahimi, Ph.D

Academic Editor

PLOS ONE

Journal Requirements:

Reviewers' comments:

Reviewer's Responses to Questions

**Comments to the Author**

1. Is the manuscript technically sound, and do the data support the conclusions?

Reviewer #1: Yes

Reviewer #2: Yes

2. Has the statistical analysis been performed appropriately and rigorously? 

Reviewer #1: Yes

Reviewer #2: Yes

3. Have the authors made all data underlying the findings in their manuscript fully available?

Reviewer #1: Yes

Reviewer #2: Yes

4. Is the manuscript presented in an intelligible fashion and written in standard English?

Reviewer #1: Yes

Reviewer #2: Yes

5. Review Comments to the Author

Reviewer #1: In this article, authors presented a systematic and narrative review of the machine learning techniques, data sources and modalities used for detection and diagnosis of cognitive impairment in Parkinson's Disease. 70 studies were included in this review, with the most relevant information extracted from each. From each study, strategy, modalities, sources, methods and outcomes were extracted. This paper clearly provides important information for the versatility of machine learning in analysing a range of modalities for the detection and diagnosis of cognitive impairment in Parkinson's Disease, including imaging, EEG, speech and more, yielding notable diagnostic accuracy.

Major comments:

1. Considering the nature of systematic and narrative review, please cite the following papers, presenting the association between quantitative susceptibility mapping and cognition in Parkinson’s disease in the Introduction:

10.1002/mds.27717

10.1002/mds.28077

10.3389/fnins.2022.938092

2. In Figure 2, the characters are too small to recognize the words.

Reviewer #2: Dear Authors,

your study focuses on a systematic literature review that aims to test the potential of artificial intelligence algorithms in a highly prominent clinical area, the prediction and diagnosis of non-motor symptomatology in Parkinson's Disease. Although I consider the topic to be of high interest and in accordance with the direction of today's literature, some points need to be considered to improve your work.

1. In the Abstract Background section the aim of the study should be clarified.

2. In the Abstract Results section, please change "a range of modalities..." in " a wide range of different data modalities..".

3. In the Introduction section in line 2 please change "the most common manifestation of Parkinsonism" in "the most common neurodegenerative disorder", and in line 3 change "unintended, uncontrollable movements" in "dyskinesia".

4. This point refers to the paragraph on Parkinson's disease in the Background section. Although the clinical content reported is correct, the message from the text is that this disease is primarily cognitive, as if it was a dementia, as opposed to Parkinson's being a motor disease.

Therefore, I recommend that you rewrite this section emphasizing the concept that Parkinson's is a neurodegenerative motor disease. It is important to emphasize that the manifestations at onset are motor in nature, stressing which are the main motor symptoms (rigidity, tremor, and bradykinesia), leaving as the last description the non-motor symptomatology as a manifestation both cognitive, related to the domain of execution functions, attention, visuospatial abilities, and language, given the fact that dopamine is also related to the cognitive circuit, but also anosmia, sialorrhea, and swallowing.

5. In the Background Cognitive Impairment section, after lines 96-98 in which you describe neuropsychological screening tools, I recommend that you add a sentence that tells the higher diagnostic power of MoCA than MMSE, because the first must be used in clinical practice for cognitive screening in PD.

6. In the Background Machine Learning section, after "ML has emerged as a valuable tool in a variety of healthcare applications, including the detection of PD" you cite a paper that uses data retrieved from clinical questionnaires and instrumental data, which are not very ecological. So I think you can add this reference to the previous one:

Palmirotta, Cinzia, et al. "Unveiling the Diagnostic Potential of Linguistic Markers in Identifying Individuals with Parkinson’s Disease through Artificial Intelligence: A Systematic Review." Brain Sciences 14.2 (2024): 137.

which demostrates how ecological language tasks could predict PD. This states how either ecological or instrumental data combined with ML can predict PD.

7. In the Background Supervised Learning section, in line 116 before ending the period, I suggest adding "to solve either regression or classification tasks." Since in this section you rightly describe supervised models associating them with regression tasks, classification tasks, or both; after the end of the sentence first mentioned and corrected, it seems more correct to me to explain both tasks, their differences, and how a regression task can become a classification one.

The regression task was explained in lines 133-136, so it could be moved by reframing the concept referring not to methods but to tasks, while the classification task should be added entirely because it was never explained.

8.In line 149, please add before the period "respectively for, classification and regression tasks."

9. In background Unsupervised Learning, in lines 187-189, you explain PCA but I think that it should be better explained, because important information are missing.

10. In the Background: Deep Learning section in lines 207-209 state that deep learning is so called because the models consist of multiple layers. Although deep learning models are more complex in terms of architecture, as you defined, the term deep is associated with the ability of such models to extract trainable, hierarchically organized features directly from raw data, thus not requiring the feature extractor block, i.e., engineered features extracted upstream and not trainable, before the definition of the model itself. For this reason, the architecture is more complex, i.e. formed by multiple layers. Please, rewrite this definition.

11. In lines 216-218, when you explain the limitations of deep learning models, I think that the need for very huge datasets for training is worth mentioning.

12. In the Background Ensemble learning, in lines 220-225 you explain ensemble learning benefits, the overfitting reduction is worth mentioning.

13. In Table 1, the LR symbol is associated to Logistic Regression, as I understand from the description, but in the Background supervised learning section it is associated to Linear Regression. This is ambiguous, please clarify this symbol association.

14. In the Search strategy section, Table S1 is mentioned. In Pubmed, strategy reviews are not included, what for IEEE xplore, Scopus and Science Direct? Reviews removal is not mentioned either in Figure 1 or in the text.

15. In the Data Extraction section, you describe the information obtained from papers, you mentioned Data Source but this information is not present in Table S3. Is it a typos?

Furthermore, I think that you should also extract Validation-and-testing method, ie. crossvalitation? Nested-crossavalisation? Train test split? and add it an column in Table S3.

16. I recommend to place Literature Review Eligibility section before Data Sources section, because it belongs to Observations and Findings.

17. Figure 2 is not described or mentioned in the text, only from Line 390 but is in brackets.

18. Lines 405-407, should be placed at the start of the results section.

19. Figure 3 is not described in the text.

20. Please don't cite figures in brackets.

21. Figure's quality should be improved.

22. A legend after Table S3 should be added.

23. In Data sources, Study Activities, Data Modalities, Machine laerning techiniques, Performance Metrics sections after indicating the number of papers (N=value) next to each result, the references of all papers associated with that result should be included.

24. I suggest adding another supplementary Table in which you describe Data Modalities, i.e. by putting along rows the different modalities and in the next columns all kinds of data belonging to that category. The reference in the text could be placed in Data Extraction, indicating a description of your data modality.

25. In the Results Supervised Learning section, add references to papers contributing to that result after (N=value).

26. In the Results Classification section please, before every reference, add Authors et al. not only the citation.

27. AUC and accuracy metrics are sometimes presented as percentages and other times as float numbers; please always use the same wording in the textual results section as in Table S3.

28. In line 826, what is "Per Fig.3"? Is it a typo?

29. In the Discussion section before starting, I suggest resuming the purpose of the work and a brief recap of the most important findings so that we can then tie the discussion together.

30. In line 902, you can also add the benefit of fine tuning in brief.

31. In supporting information Table S2 and Table S3 are not mentioned.

32. Please check for any typos and English grammar.

6. PLOS authors have the option to publish the peer review history of their article (what does this mean?). If published, this will include your full peer review and any attached files.

Reviewer #1: **Yes: **Yuto Uchida

Reviewer #2: No

---

## [Author Response · Author response to Decision Letter 0]

25 Apr 2024

Dear Editors and Reviewers,

We would like to thank the reviewers for their detailed and constructive feedback. Our responses to the reviewers’ comments are provided in the attached "response-to-review.pdf" file along with additional comments provided in our cover letter. The corresponding modified text is highlighted in the track changes version of the manuscript.

---

## [Decision Letter · Decision Letter 1]

30 Apr 2024

Machine learning for the detection and diagnosis of cognitive impairment in Parkinson’s disease: a systematic review

PONE-D-24-09980R1

Dear Dr. Altham,

We’re pleased to inform you that your manuscript has been judged scientifically suitable for publication and will be formally accepted for publication once it meets all outstanding technical requirements.

Kind regards,

Farzin Hajebrahimi, Ph.D

Academic Editor

PLOS ONE

Reviewers' comments:

Reviewer's Responses to Questions

**Comments to the Author**

1. If the authors have adequately addressed your comments raised in a previous round of review and you feel that this manuscript is now acceptable for publication, you may indicate that here to bypass the “Comments to the Author” section, enter your conflict of interest statement in the “Confidential to Editor” section, and submit your "Accept" recommendation.

Reviewer #1: All comments have been addressed

Reviewer #2: All comments have been addressed

2. Is the manuscript technically sound, and do the data support the conclusions?

Reviewer #1: Yes

Reviewer #2: Yes

3. Has the statistical analysis been performed appropriately and rigorously? 

Reviewer #1: Yes

Reviewer #2: Yes

4. Have the authors made all data underlying the findings in their manuscript fully available?

Reviewer #1: Yes

Reviewer #2: Yes

5. Is the manuscript presented in an intelligible fashion and written in standard English?

Reviewer #1: Yes

Reviewer #2: Yes

6. Review Comments to the Author

Reviewer #1: (No Response)

Reviewer #2: (No Response)

7. PLOS authors have the option to publish the peer review history of their article (what does this mean?). If published, this will include your full peer review and any attached files.

Reviewer #1: **Yes: **Yuto Uchida

Reviewer #2: No

---

## [Editor Report · Acceptance letter]

2 May 2024

PONE-D-24-09980R1 

PLOS ONE

Dear Dr. Altham, 

I'm pleased to inform you that your manuscript has been deemed suitable for publication in PLOS ONE. Congratulations! Your manuscript is now being handed over to our production team.

Kind regards, 

on behalf of

Dr. Farzin Hajebrahimi 

Academic Editor

PLOS ONE